# ARGen-Dexion: Autoregressive Image Generation Made Stronger by Vision Decoder

## Abstract

Autoregressive models (ARGen) have emerged as a cornerstone for image generation within multimodal large language models (MLLMs), yet their visual outputs remain stubbornly underwhelming. Traditional efforts, scaling AR models or re-engineering architectures, yield diminishing returns at exorbitant cost, straining infrastructure without resolving core limitations. In this work, we challenge the status quo, asserting that vision decoders must shoulder greater responsibility for image synthesis, liberating autoregressive models from undue burden. We present ARGen-Dexion, a systematic overhaul of the vision decoder that redefines autoregressive image generation without modifying pre-trained AR models or visual encoders. Our approach delivers transformative gains through three innovations: (1) a scaled, fine-tuned decoder achieving unprecedented reconstruction fidelity, (2) bi-directional Transformer-based token refiner that infuses global context to refine the AR model outputs, shattering the constraints of causal inference inherent, and (3) a resolution-aware training strategy enabling seamless multi-resolution and multi-aspect-ratio synthesis. Extensive scaling studies unveil deep insights into decoder design, challenging long-held assumptions. Empirically, ARGen-Dexion boosts LlamaGen by a striking 9% VQAScore on the GenAI-Benchmark and 4% GenEval performance. Moreover, it can be applied to various discrete MLLMs. This work compels a bold rethinking of the interplay between MLLMs and vision decoders, paving the way for efficient and visually superior multimodal systems.

## 1 Introduction

The "next-token-prediction" paradigm has become the *de facto* standard for large language models (LLMs) (Touvron et al., 2023a;b; Dubey et al., 2024). In recent years, it is gaining increasing popularity in image generation, driven by the ambition of multimodal large language models (MLLMs).

Leveraging discrete visual tokenizers, autoregressive (AR) models sequentially predict visual tokens to generate images, mirroring the process of text generation — a paradigm we term ARGen. Despite its promise, ARGen lags behind state-of-the-art (SOTA) diffusion models (Dai et al., 2023; Polyak et al., 2024), primarily due to two intrinsic limitations: suboptimal visual tokenization and the constraints of causal inference (Zhou et al., 2024; Fan et al., 2024; Li et al., 2024b). Discrete visual tokenizers inherently lose fine-grained information during quantization, capping the fidelity of reconstructed images. Meanwhile, causal inference enforces a unidirectional flow of information, overlooking the global coherence for vision. These limitations have sparked a surge of research to elevate ARGen's performance. Some strategies tackle these issues by introducing additional complexity to large language models (LLMs) — incorporating auxiliary training losses (Li et al., 2024b; Sun et al., 2024c), modifying causal masking (Tian et al., 2024), or adopting novel training regimes (Yu et al., 2024; Pang et al., 2024). While these solutions yield performance gains, they often come at the cost of increased engineering complexity, scalability bottlenecks, and potential trade-offs with MLLMs capabilities. Conversely, more straightforward approaches adhere to the classic "next-token-prediction" framework, scaling AR models and training resources (Wang et al., 2024; Team, 2024) to chase incremental improvements. While these methods are more elastic, the marginal gains often fail to justify the steep computational costs. These challenges underscore the need for a new perspective on AR-based image generation — one that harmonizes tokenization fidelity, bidirectional context modeling, and computational efficiency.

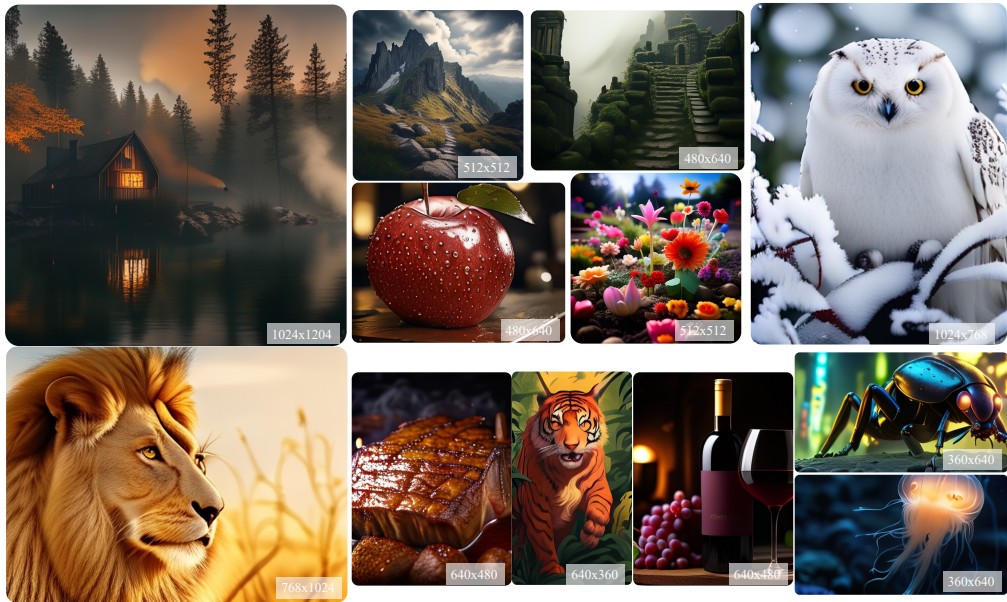

Figure 1: ARGen-Dexion enhances image generation with pre-trained AR models. The showcased images are generated using a 0.8B LlamaGen model augmented by ARGen-Dexion, enabling high quality generation with various resolutions (which is unavailable in vanilla LlamaGen).

In contrast to prior methods, we pose a pivotal question: Can we enhance image generation quality without introducing additional complexities to AR models? To address this, we shift our focus to the vision decoder in ARGen — the critical final stage of image synthesis. ***We have the postulation that the decoder should extend beyond mere reconstruction to also participate in generation***. This insight suggests that the vision decoder can share the generative responsibility, reducing the burden on AR models and fostering a more balanced, effective image synthesis pipeline.

To realize this vision, we propose ARGen-Dexion, a next-generation decoder for autoregressive image generation. First, we investigate the scaling law of the decoder to enhance reconstruction performance. By jointly training with a pre-trained, frozen vision encoder using reconstruction loss, we show that the decoder strictly follows scaling laws in relation to model size and training cost, effectively mitigating the reconstruction limitations of discrete image tokenizers. Next, we introduce a token refiner within the decoder — a module built with bi-directional Transformer blocks. Trained with cross-entropy loss, the token refiner substantially reduces accumulated visual token prediction errors caused by causal inference, further elevating image synthesis quality. Moreover, we incorporate multi-resolution generation directly within the decoder, eliminating the need to train AR models on extra images of varying resolutions.

ARGen-Dexion clearly enhances the performance of pretrained ARGen models without additional overhead. Built upon the pretrained LlamaGen, our approach boosts the performance from 0.59 to 0.68 on GenAI-Bench (Li et al., 2024a), and from 0.32 to 0.36 on GenEval (Ghosh et al., 2024). Notably, ARGen-Dexion seamlessly integrates with all MLLMs within the "next-token-prediction" framework for image generation, including models like Lumina-mGPT (Liu et al., 2024) and EMU3 (Wang et al., 2024). Our exploration of ARGen-Dexion underscores the immense potential of vision decoders, unveiling a vast and untapped frontier of possibilities for autoregressive image synthesis.

## 2 RELATED WORK

### 2.1 IMAGE GENERATION WITH AUTOREGRESSIVE

Image generation has witnessed rapid advancements recently, primarily driven by the evolution of Diffusion models (Lipman et al., 2022; Dai et al., 2023; Polyak et al., 2024). Beyond the Diffusion paradigm, autoregressive models have also emerged as a compelling approach, which typically relies on a discrete visual tokenizer (*e.g.*, VQGAN (Esser et al., 2021), VQVAE (Van Den Oord et al.,

2017), *etc.*) to transform images into discrete tokens. An autoregressive model is then employed to learn these visual tokens using a "next-token-prediction" paradigm, akin to the pipeline of large language models. Subsequently, a discrete visual decoder reconstructs the generated visual tokens into an image. We refer to this pipeline as ARGen for brevity. A notable example of ARGen is LlamaGen (Sun et al., 2024a), which epitomizes simplicity within this framework while delivering impressive results. By incorporating text embeddings, LlamaGen achieves causal prediction of visual tokens guided by textual input. Similarly, EMU3 (Wang et al., 2024) and Chameleon (Team, 2024) adhere to the traditional ARGen pipeline while preserving the text generation capabilities inherent in autoregressive large language models, thereby paving the way for multimodal large language models. Additionally, several exploratory efforts have sought to improve generation quality within the ARGen framework by rethinking its core design, like VAR (Tian et al., 2024) and MAR (Li et al., 2024b). Further, Fluid (Fan et al., 2024) delves into the impact of visual representation (continuous or discrete) and token order (causal or random), offering deeper insights into these dimensions. Adhering to the principle of simplicity, our work avoids altering AR models. Instead, we focus on optimizing the standalone visual decoder for ARGen.

## 2.2 TOWARDS MLLMS WITH IMAGE GENERATION

A strong motivation for ARGen is the ambition to equip MLLMs with advanced image generation capabilities. Without altering the structure of large language models (LLMs), some methods (like Chameleon (Team, 2024), Lumina-mGPT (Liu et al., 2024), and EMU3 (Wang et al., 2024)) achieve impressive generation quality. Conversely, some approaches prioritize improved ARGen quality at the cost of increased complexity in LLM implementation, like EMU (Sun et al., 2024c), EMU2 (Sun et al., 2024b), and SEED-X (Ge et al., 2024), which require additional regression losses for visual features during LLM training. Similarly, Transfusion (Zhou et al., 2024) leverages Diffusion loss and bidirectional interactions by modifying causal masking. Other efforts, such as LlamaFusion (Shi et al., 2024), MoMa (Lin et al., 2024), and Libra (Xu et al., 2024), explore mixture-of-experts (MOE) or modality-specific feedforward layers for visual features. Instead of focusing on LLMs and introducing additional engineering and infrastructure hurdles, we redirect our attention to the visual decoder, the final stage of image generation in unified MLLMs.

## 2.3 VISUAL TOKENIZER

Several primary directions have been explored to enhance the visual tokenizer for MLLMs. First, a widely adopted approach involves training VQGAN architectures with larger and higher-quality datasets or employing carefully designed hyperparameters to improve representational capabilities, as demonstrated by LlamaGen (Sun et al., 2024a) and Chameleon (Team, 2024). Second, innovative designs for visual tokenizers have been proposed. For instance, VAR (Tian et al., 2024) employs residual designs for improved reconstruction, while MAGVIT-v2 (Yu et al., 2023) introduces lookup-free quantization (LFQ) to enable learning of a larger vocabulary. Finally, a critical challenge in MLLMs lies in aligning semantic-level and pixel-level representations for tasks such as understanding and generation. Exploratory works like VILA-U (Wu et al., 2024b) and TokenFlow (Qu et al., 2024) offer profound insights into this alignment issue. In contrast to prior efforts that address both encoder and decoder design, our work focuses on the standalone decoder for better performance.

# 3 ARGEN-DEXION

## 3.1 IMAGE GENERATION WITH AUTOREGRESSIVE MODELS

Given an input condition $c$ and the previous visual tokens $\{x_1, x_2, \cdots, x_{t-1}\}$ AR models predict the conditional probability distribution of the next image token $x_t$ as:

$$P\left(x_t \mid c, x_1, x_2, \ldots, x_{t-1}\right). \tag{1}$$

By sequentially generating, an image can be synthesized with a discrete image decoder conditioned on the input. While simple and effective, two key limitations hinder generation quality. First, Eq. 1 models the probability of the next token, relying on a discrete representation. This necessitates the use of an image tokenizer with vector quantization, which introduces information loss and constrains reconstruction quality. Second, the causal masking applied in attention mechanisms significantly

limits bi-directional interactions, which are crucial for capturing comprehensive image context. These challenges have been widely recognized, with extensive research addressing these issues (Zhou et al., 2024; Li et al., 2024b).Notably, Fluid (Fan et al., 2024) offers a detailed analysis of these limitations. Unlike the aforementioned methods, we address these two issues within the image decoder, avoiding imposing additional infrastructure and engineering challenges while significantly reducing computational requirements compared to scaling MLLMs.

## 3.2 ARGEN-DEXION OVERVIEW

Our goal is to design a simple yet effective image decoder for ARGen to address the aforementioned limitations. ARGen-Dexion achieves this simplicity with two components: a multi-stage main vision decoder capable of generating images at arbitrary resolutions and aspect ratios, and a multi-layer Transformer-based token refiner that iteratively alters the AR model generated tokens. Fig. 2 illustrates the design. During inference, ARGen-Dexion first refines the tokens generated by the AR model using the token refiner, and the refined tokens are then decoded by the main decoder to produce the final image.

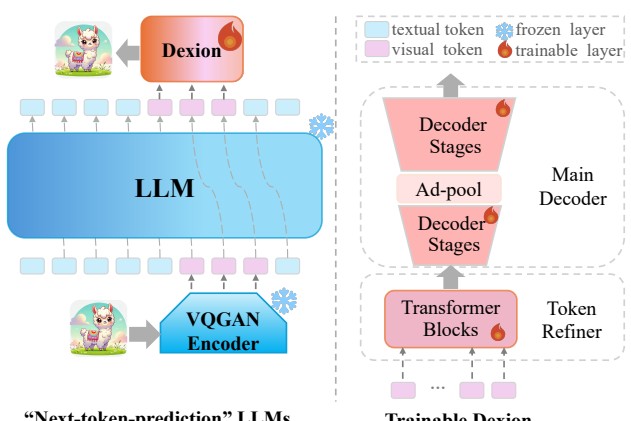

Figure 2: ARGen-Dexion pipeline. **Left** shows the pipeline of ARGen, and **right** shows the design of our Dexion for ARGen. Please check Sec. 3 for details and training.

In detail, our token refiner comprises multiple vanilla bi-directional Transformer blocks to refine input tokens, following the operations in MaskGIT (Chang et al., 2022). In the main decoder, the first stage uses Transformer blocks to capture global interactions, while the remaining stages leverage ConvNeXt (Liu et al., 2022) blocks to extract local features. We configure all attention blocks with 8 heads, ConvNeXt blocks with a kernel size of 7, and an MLP expansion ratio of 4. An adaptive pooling layer is introduced after the second stage to enable generation at arbitrary resolutions and aspect ratios. ARGen-Dexion features a standalone codebook separate from the encoder, offering flexibility in dimensionality. This allows for scaling both the depth and channel dimensions of ARGen-Dexion. While further optimization of hyper-parameters could enhance performance, it lies beyond the scope of this work.

## 3.3 TRAIN DEXION

Dexion architecture is both elegant and simple, it employs a two-step training strategy: first, the main decoder is trained to enhance reconstruction with reconstruction loss. Then we train the refiner with cross-entropy loss to refine the output tokens generated by the AR model. For illustration, we build our model using a pretrained LlamaGen AR model and corresponding VQGAN model.

**Train for reconstruction** To align with the pretrained LlamaGen T2I task, we first resize the input image $x$ to a resolution of $512 \times 512$, unless otherwise specified, and utilize the pretrained VQGAN encoder with the provided T2I checkpoint to convert the image into tokens. Using these encoded tokens, we index the codebook in Dexion to convert them into a feature map, which is then decoded into pixel space $\hat{x}$. Following LlamaGen, we optimize the main decoder with:

$$\mathcal{L} = \mathcal{L}_2(x, \hat{x}) + \lambda_{\mathrm{p}} \mathcal{L}_{\mathrm{P}}(x, \hat{x}) + \lambda_{\mathrm{G}} \mathcal{L}_{\mathrm{G}}(\hat{x}), \tag{2}$$

where $\mathcal{L}_2(\cdot)$ indicates L2 reconstruction loss, $\mathcal{L}_{\mathrm{P}}(\cdot)$ represents LPIPS perceptual loss (Zhang et al., 2018), and $\mathcal{L}_{\mathrm{G}}(\cdot)$ is the adversarial loss (Isola et al., 2017). We use $\lambda_{\mathrm{p}}$ and $\lambda_{\mathrm{G}}$ to balance the losses.

**Train for Token-Refiner** Another key component of ARGen-Dexion is the Token Refiner, which addresses the limitations of causal inference in AR models, as discussed in Fluid (Fan et al., 2024). Rather than complicating the AR model itself, we offload part of the task to the decoder. We allow AR model to generate imperfect tokens, and augment with bi-directional Transformer blocks for

globally coherent visual representations. Drawing inspiration from the MaskGIT (Chang et al., 2022) pipeline, this approach enhances flexibility and efficiency.

We begin with image tokens extracted using the VQGAN encoder and randomly replace a portion of the ground truth tokens with [mask] tokens, using a masking ratio between $[0.001, 0.3]$ to simulate the refinement of partially generated images rather than generating them from scratch. The entire token sequence, including the masked tokens, is processed by

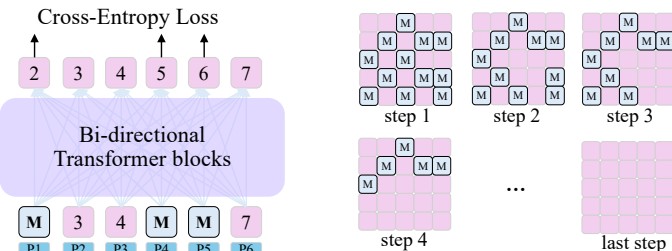

Figure 3: Left indicates the training of our MaskGIT-based refiner, and right shows examples of the iteratively refining.

the refiner, which predicts the original tokens. Training is driven by cross-entropy loss, applied exclusively to the masked tokens, guiding the refiner to accurately restore the masked or noisy tokens.

During inference, we identify the lowest-confidence tokens from the AR model's output and replace them with [mask] tokens, aiming to refine the generation without heavily altering the original content. The refiner then iteratively predicts these masked tokens, progressively filling in predictions based on confidence scores. This iterative process continues until all masked tokens are resolved, with detailed analysis of different inference schedules provided in the experimental section.

### 3.4 MULTI-RESOLUTION IMAGE GENERATION

Multi-resolution and multi-aspect-ratio image generation, as demonstrated in EMU3 (Wang et al., 2024) and Lumina-mGPT (Liu et al., 2024), typically involves training MLLMs on image data with diverse resolutions. However, this approach incurs high training costs and often leads to suboptimal generation quality. To overcome this, we shift the complexity to the decoder. Instead of training MLLMs on multiple resolutions, we train MLLMs at a single resolution, and enable the decoder to generate images at arbitrary (pre-defined) resolutions based on a given resolution hyper-parameter.

We introduce an adaptive-pooling layer after the second stage of the main decoder, fine-tuning it to handle specific resolutions. Although a more advanced design could potentially improve results, this is not our primary focus. Our experiments show that this approach effectively generates images with the desired resolution and aspect ratio, all while avoiding additional complexity for the MLLMs.

## 4 EXPERIMENTS

### 4.1 DATASET AND EVALUATION

Training ARGen-Dexion is both data and computationally efficient compared to training or fine-tuning an AR model for image generation. Unlike traditional AR models, ARGen-Dexion only requires images, without the need for captions or prompts. Most of our scaling experiments are conducted on the ImageNet dataset, while fine-tuning utilizes a curated set of high-aesthetic-quality images.

To evaluate reconstruction quality, we report standard metrics such as PSNR, SSIM, and rFID on the ImageNet validation set. For assessing the token refiner's performance, we examine the training loss and generation results on GenAI-Bench (Li et al., 2024a). Additionally, we provide detailed results on both GenAI-Bench and GenEval (Ghosh et al., 2024) to assess generation quality. The detailed training setups are presented in supplementary.

### 4.2 SCALING DECODER FOR RECONSTRUCTION

We begin by conducting experiments on reconstruction to evaluate the performance of the scaled Dexion main decoder, while excluding the Token Refiner module. Specifically, we explore the scaling laws from three perspectives: training cost, training data size, and model size. By default, we utilize the VQGAN encoder to extract tokens and train ARGen-Dexion at a resolution of 256.

**Training cost Scaling** We first examine the impact of training cost on reconstruction performance. A common practice in training VQGAN is to introduce $\lambda_G$ after a certain number of iterations, which complicates reliable analysis. Therefore, to investigate the effect of training cost, we focus exclusively on the L2 reconstruction loss $\ell_2(x, \hat{x})$ and

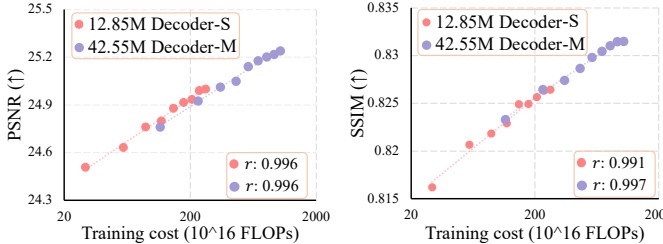

Figure 4: **Scaling training costs** can consistently improve reconstruction. "$r$" indicates Pearson correlation coefficients.

LPIPS perceptual loss $\mathcal{L}_P(x, \hat{x})$ throughout the training process. Reconstruction performance is evaluated using PSNR and SSIM. Experimental results in Fig. 4 show that *reconstruction performance of decoder consistently improves with increased training cost, strictly following the scaling law*, with a Pearson correlation coefficient $r > 0.99$, indicating a strong correlation. That is, higher training cost results in better and predictably improved reconstruction performance.

**Data size Scaling** Next, we investigate the impact of training data size on reconstruction performance. We use 10%, 20%, 40%, 60%, 80%, 100% of the ImageNet training set, keeping the number of training iterations constant across all settings. The adversarial loss $\mathcal{L}_G(\cdot)$ is introduced after $20k$ iterations, and we disable the learning rate scheduler

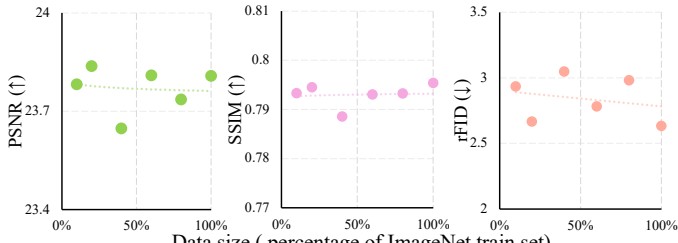

Figure 5: **Scaling training data** cannot improve reconstruction.

to eliminate the influence of varying epoch counts. Interestingly, as shown in Fig. 5, even a small dataset can achieve performance comparable to one that is $10\times$ larger. This suggests that *increasing the training data size does not always translate to better reconstruction quality*.

**Model size Scaling** Lastly, we examine the impact of model size on reconstruction performance. We scale the main decoder from 4M to 166M parameters, with detailed configurations provided in Table 1. All models are trained on the full ImageNet training set for 40 epochs.

As shown in Fig. 6, PSNR, SSIM, and rFID results are plotted against model size and FLOPs for single-image inference. The results reveal that *PSNR and SSIM consistently improve with increasing model size*, with a Pearson correlation coefficient $r > 0.9$. In contrast, *rFID saturates once the decoder size exceeds after 50M parameters*,

| | Params (M) | FLOPs (G) | Dimensions |
|---|---|---|---|
| Dexion-XXS | 4.3 | 16.2 | [128, 96, 64, 48, 32] |
| Dexion-XS | 7.6 | 31.6 | [192, 128, 96, 64, 48] |
| Dexion-S | 12.9 | 61.5 | [256, 192, 128, 96, 64] |
| Dexion-B | 23.7 | 121.7 | [384, 256, 192, 128, 96] |
| Dexion-M | 42.6 | 240.0 | [512, 384, 256, 192, 128] |
| Dexion-L | 81.7 | 477.6 | [768, 512, 384, 256, 192] |
| Dexion-XL | 166.5 | 968.8 | [1152, 768, 512, 384, 256] |

Table 1: Dexion decoder scales (4M–166M parameters) with a fixed block configuration of $[3, 9, 6, 3, 3]$ across five stages.

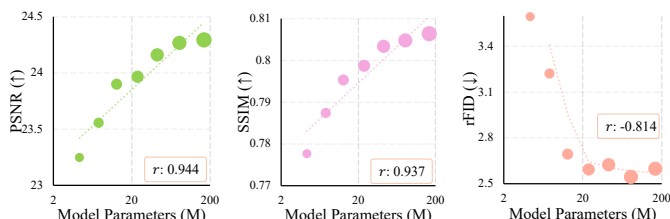

Figure 6: **Scaling model size** consistently improves reconstruction in PSNR and SSIM but saturates with rFID.

suggesting diminishing returns for perceptual quality at larger decoders.

**Reconstruction vs. Generation** Reconstruction aids generation, but they are distinct tasks. To assess whether better reconstruction implies better generation, we decode fixed latent tokens using checkpoints from model scaling experiments and evaluate VQAScore on GenAI-Bench (Li et al., 2024a). Fig. 7 presents results for "Basic" and "Hard" prompts. Meanwhile, we also report the

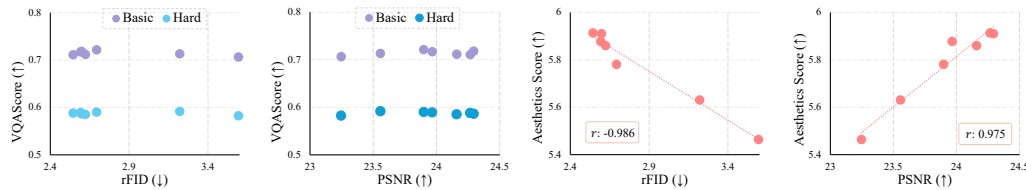

Figure 7: **Improved reconstruction** does not imply superior generation (measured by VQAScore), but superior aesthetics score.

| Method | ratio | Codebook | rFID(↓) | PSNR(↑) | SSIM(↑) |
|--------|-------|----------|---------|---------|---------|
| VQGAN (Esser et al., 2021) | 16 | 16384 | 4.99 | 20.00 | 0.629 |
| MaskGIT (Chang et al., 2022) | 16 | 1024 | 2.28 | - | - |
| LlamaGen (Sun et al., 2024a) | 16 | 16384 | 2.19 | 20.79 | 0.675 |
| LlamaGen-**Dexion** (42M) | 16 | 16384 | 2.62 | 24.16 | 0.803 |
| LlamaGen-**Dexion** (166M) | 16 | 16384 | 2.60 | 24.29 | 0.806 |

Table 2: Comparisons with other discrete tokenizers. We train on ImageNet train set and evaluate on 256×256 50k validation set. For our Dexion, we use LlamaGen pretrained VQGAN T2I encoder.

Aesthetics Score (AS) (Schuhmann et al., 2022) for generated images to evaluate the image aesthetics. As shown in Fig.7 (top row), our results indicate that ***better reconstruction quality does not always lead to improved image generation quality measured by metrics like VQAScore***. This reinforces the distinction between reconstruction and generation, underscoring the need for approaches beyond simple reconstruction optimization. However, as highlighted in Fig.7 (bottom row), ***better reconstruction consistently enhances aesthetics scores***, demonstrating its role in refining visual details and fidelity.

### 4.3 MAIN DECODER DESIGN CHOICE

As analyzed before, we observe that increased training cost and larger model size consistently improve reconstruction quality. However, these improvements are not significant. For instance, with nearly $10\times$ more training cost, PSNR increases marginally from 25.0 to 25.2, and SSIM improves slightly from 0.828 to 0.831. Similarly, scaling the model size by $4\times$ results in only minor gains in PSNR and SSIM, with almost no reduction in rFID. Meanwhile, experiments indicate that improved reconstruction quality does not imply better generation results. Given the computational limitations, all subsequent experiments will be conducted using the 42M Dexion model. Table 2 provides a detailed comparison with other image tokenizers under a fair comparison.

For image generation, we further fine-tune our 42M-parameter Dexion model on the ImageNet dataset to reconstruct images at a resolution of 512×512. We then fine-tune this pretrained model on a curated dataset of 2 million licensed and synthetic high-aesthetic-quality images. To enhance performance during fine-tuning, we lower the initial learning rate from 3e-4 to 3e-5.

### 4.4 SCALING REFINER FOR REFINEMENT

As mentioned earlier, we introduce the Token Refiner to subtly enhance the predicted tokens, refining the generated output without significantly altering the overall context. Next, we explore the scaling properties of the Token Refiner in Dexion, evaluating three model sizes — Small, Base, and Large — ranging from 52M to 225M parameters. To facilitate token refinement, we incorporate an additional classification head to predict the refined tokens, with detailed configurations provided in Table 3. Treating the classification task as a form of pretraining, we present the relationship between training FLOPs and training NLL in Figure 8.

As shown in Fig. 8, the Token Refiner design follows the scaling law with respect to training cost, demonstrating high Pearson correlation coefficients. Notably, we find that a medium-sized refiner already delivers robust performance, while scaling to larger models yields diminishing returns. While more advanced training strategies may improve the performance of larger refiners, exploring this potential lies beyond the scope of our current work. Based on these findings, we select the Base version of the Refiner for subsequent experiments. We then investigate key aspects of the inference process, including masking ratios, scheduling strategies, and the number of inference steps.

| Refiner | S | M | L |
|---|---|---|---|
| Depth | 12 | 12 | 16 |
| Heads | 8 | 8 | 8 |
| MLP expand | 4 | 4 | 4 |
| Width | 512 | 768 | 1024 |
| Params(M) | 52.6 | 105.9 | 225.2 |
| FLOPs(G) | 60.2 | 119.3 | 257.9 |

Table 3: Configurations of different Refiner design sizes.

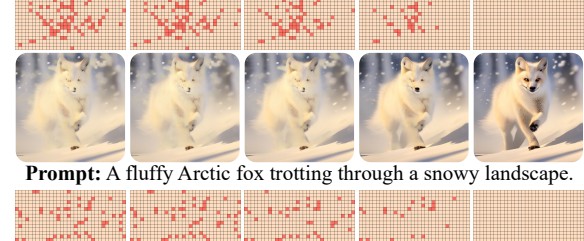

Figure 8: Scaling Refiner.

| Schedule | VQAScore |
|---|---|
| Linear | 0.660 |
| Cosine | **0.662** |
| Sqrt | 0.659 |
| Identity | **0.662** |

Table 6: Different schedulers perform similar.

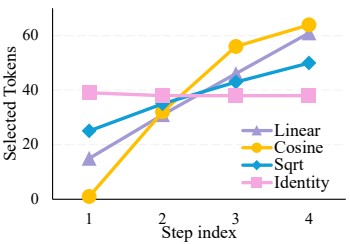

Figure 9: Inference Scheduler.

**Prompt:** A fluffy Arctic fox trotting through a snowy landscape.

**Prompt:** A peaceful alpine valley with a meandering stream, wildflowers scattered across the grass

Figure 10: Token refiner progressively infers masked tokens for better results. Red blocks represent masked tokens, yellow blocks indicate unmasked tokens.

## 4.5 REFINER INFERENCE STUDY

The inference process of the Token Refiner involves several hyper-parameters that may influence both performance and efficiency, similar to MaskGIT. In this subsection, we delve into these factors, systematically exploring different configurations to determine the optimal inference settings. For a fair comparison, we use identical AR model outputs for each study.

**Study Refiner Masking ratio during inference** We first investigate the impact of the masking ratio, which controls the proportion of tokens masked during inference. For simplicity, we fix the number of inference steps to 8 and adopt a cosine masking schedule. Considering the masking ratio is randomly sampled from the range $[0.001, 0.3]$ during training, we evaluate performance at

| Mask Ratio | 0.05 | 0.1 | 0.15 | 0.2 | 0.25 | 0.3 |
|---|---|---|---|---|---|---|
| VQAScore | 0.662 | 0.661 | **0.663** | 0.661 | 0.658 | 0.655 |

Table 4: A relatively smaller refiner masking ratio during inference would not impact the overall hard-prompt VQAScore, but larger masking ratio leads to slightly worse performance.

| Inference steps | 4 | 8 | 12 | 16 | 20 |
|---|---|---|---|---|---|
| VQAScore | **0.675** | 0.674 | 0.671 | 0.671 | 0.673 |

Table 5: Our refiner is not sensitive to inference steps.

$[0.05, 0.1, 0.15, 0.2, 0.25, 0.3]$ masking ratios to assess the effect on generation quality. The generation results are assessed using GenAI-Bench, and the overall VQAScore is reported in Table 4. Our results show that a relatively small masking ratio is sufficient to achieve a high VQAScore. However, as the masking ratio increases, we observe a slight decline in performance. Based on these findings, we set the masking ratio to 0.15 for all subsequent experiments.

**Study Refiner Inference Step** Next, we investigate the effect of inference steps. We tested a range of step counts, from 4 to 20, with the results summarized in Table 5. Interestingly, in contrast to

| Model | Type | # Params | GenEval. | GenAI-Bench | |
| --- | --- | --- | --- | --- | --- |
| | | | | Basic | Advanced |
| SDv2.1 (Rombach et al., 2022) | Diff. | 0.9B | 0.50 | 0.78 | 0.62 |
| SDXL (Podell et al., 2023) | Diff. | 2.6B | 0.55 | 0.84 | 0.63 |
| Show-o (Xie et al., 2024) | AR.+Diff. | 1.3B | 0.53 | 0.70 | 0.60 |
| SEED-X (Ge et al., 2024) | AR.+Diff. | 17B | 0.49 | 0.86 | 0.70 |
| EMU3 (Wang et al., 2024) | AR. | 8B | 0.66 | 0.78 | 0.60 |
| LlamaGen (Sun et al., 2024a) | AR. | 0.8B | 0.32 | 0.74 | 0.59 |
| LlamaGen-**Dexion** | AR. | 0.8B | 0.36 (↑ 0.4) | 0.74(↑ 0.0) | 0.68(↑ 0.9) |
| Janus-Pro (Chen et al., 2025) | AR. | 7B | 0.80 | 0.86 | 0.66 |
| Janus-Pro-**Dexion** | AR. | 7B | 0.81(↑ 0.1) | 0.87(↑ 0.1) | 0.74(↑ 0.8) |

Table 7: Evaluation results on the GenEval (Ghosh et al., 2024) and GenAI-Bench (Li et al., 2024a).

MaskGIT, the performance of our refiner remains stable across different step settings. This robustness can be attributed to two factors: (1) the refiner only adjusts a small subset of tokens, and (2) the generation (MaskGIT) and refinement (ours) objectives are inherently distinct. To optimize both performance and efficiency, we set the number of inference steps to 4 for following experiments.

**Study Refiner Inference schedule** As observed in MaskGIT, the choice of de-masking schedule can impact the final results. To explore this further, we experimented with several scheduling strategies, as shown in Fig. 9, and evaluated the generation quality, with results summarized in Table 6. We found that the differences across these schedulers were minimal. This is likely because our refiner operates similarly to the final steps of MaskGIT, where the influence of the scheduling on the final results is inherently limited.

**Refiner visualization** To gain deeper insight into the refinement process of our token refiner, we visualize the intermediate results in Fig. 10. For clarity, masked tokens are replaced with the token indexed as 0, allowing us to observe how the refiner progressively enhances the visual coherence and detail of the generated images through each iteration.

### 4.6 MULTI-SCALE & MULTI-ASPECT-RATIO DECODING

As discussed in Sec.3.4, we enable multi-scale and multi-aspect ratio generation directly in the decoder, rather than relying on AR models. In our experiments, we consider eight resolutions: $360 \times 640$, $480 \times 640$, $512 \times 512$, $640 \times 360$, $640 \times 480$, $768 \times 1024$, $1024 \times 768$, and $1024 \times 1024$. To implement this, we resize and center-crop a fixed-resolution image input to the encoder (e.g., $512 \times 512$ for LlamaGen) to match the target resolution, and then train the decoder to reconstruct these resized and cropped images. This approach allows the decoder to generate images in arbitrary resolutions without introducing distortions, as demonstrated in Fig. 1 and Fig.11 in the supplementary.

### 4.7 GENERATION BENCHMARK

We also report the generation evaluations on both the GenAI-Bench (Li et al., 2024a) and GenEval (Ghosh et al., 2024) benchmarks. To enhance text alignment, we apply prompt rewriting for our Dexion, following prior work in this area. The results in Table 7 show that ARGen-Dexion improves the overall evaluation performance. Full evaluation results can be found in the appendix.

## 5 CONCLUSION AND LIMITATION

Causal inference and quantization information loss are often regarded as major limitations in AR-based image generation. Instead of solely focusing on AR models, we shift our attention to the vision decoder to effectively address these challenges. In this work, we present ARGen-Dexion, an approach that effectively and efficiently tackles these issues. By scaling and optimizing the main decoder, we significantly enhance reconstruction quality, while the introduction of a token refiner improves global coherence. Additionally, Dexion enables to generate images at multiple resolutions and aspect ratios, reducing the training cost for AR models to support these capabilities. While we acknowledge ARGen-Dexion is largely constrained by the quality of predictions from the AR model (as discussed in the limitation section in the appendix), extensive experiments validate the superior performance.

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

## A APPENDIX

In the supplementary, we first discuss the limitation of ARGen-Dexion. We then detail our training recipe in Sec.C, followed by comprehensive evaluation results on GenAI-Bench and GenEval benchmarks in Sec.D. Finally, we showcase additional visual examples and more detailed token refiner studies in Sec. E.

**Large Language Models Usage** We only use extra Large Language Models to aid or polish writing. All ideas, methods, experiments, analyses, and writing were done independently by the authors.

## B LIMITATIONS

As demonstrated, the performance of ARGen-Dexion is largely constrained by the quality of predictions from the AR model. While ARGen-Dexion improves image quality, its impact is limited by its role as a decoder trained exclusively on image data. Reconstruction and generation are fundamentally different tasks. A promising future direction would be to incorporate the decoder as an integral component of generative models, enabling joint training for greater flexibility and improved results.

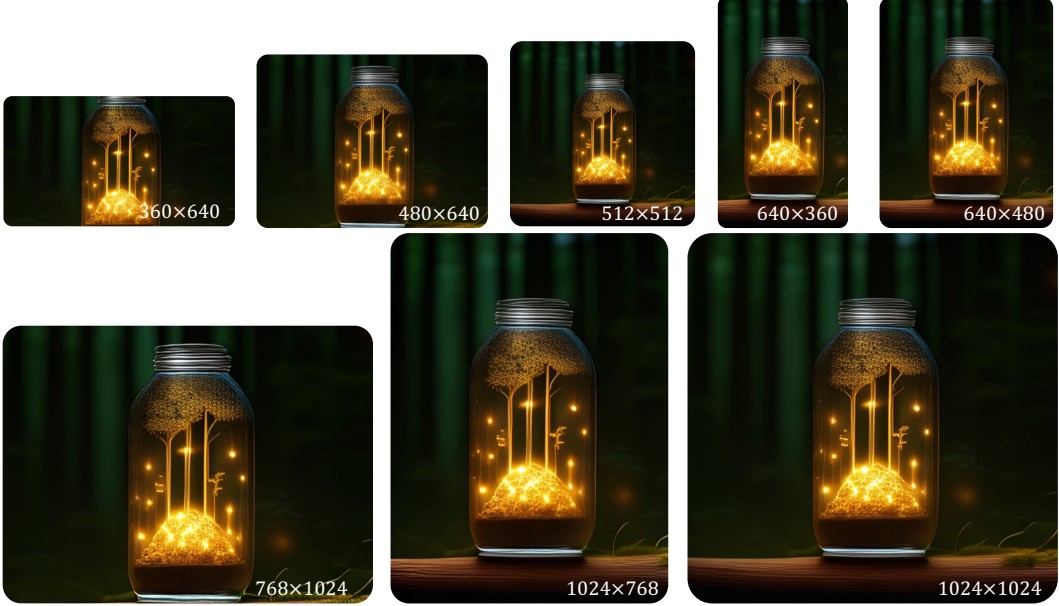

**Prompt:** A jar glows with golden light, tiny trees inside shimmer.

Figure 11: Dexion takes the responsibility to generate arbitrary resolution images conditioned on fixed AR-model outputs. This example shows the resolutions and aspect-ratios that we support. For illustration, we keep the AR output tokens identical across images.

## C TRAINING DETAILS

### C.1 TRAINING DEXION DECODER

We trained the decoder using a carefully tuned recipe to ensure stability and optimal performance. We first pre-train the model with a resolution of $256 \times 256$. The training setup utilized a learning rate of $3e-4$ with a global batch size of 128. We applied a weight decay of 0.05 and used the Adam optimizer with $\beta_1 = 0.9$ and $\beta_2 = 0.95$. To prevent exploding gradients, we set the maximum gradient norm to 1.0. Training was conducted with mixed-precision (fp16) to enhance computational efficiency.

The training objective combined multiple loss components to balance reconstruction quality and perceptual fidelity. The reconstruction loss was measured using an L2 loss. Additionally, we

| Model | Type | "Basic" prompts | | | | | | "Hard" prompts | | | | | |
| | | Attribute | Scene | Relation | | | Overall | Count | Differ | Compare | Logical | | Overall |
| | | | | Spatial | Action | Part | | | | | Negate | Universal | |
|---|---|---|---|---|---|---|---|---|---|---|---|---|---|
| SDXL-v2.1 | Diff. | 0.80 | 0.79 | 0.76 | 0.77 | 0.80 | 0.78 | 0.68 | 0.70 | 0.68 | 0.54 | 0.64 | 0.62 |
| SD-XL | Diff. | 0.84 | 0.84 | 0.82 | 0.83 | 0.89 | 0.84 | 0.71 | 0.73 | 0.69 | 0.50 | 0.66 | 0.63 |
| SD-XL Turbo | Diff. | 0.85 | 0.85 | 0.80 | 0.82 | 0.89 | 0.84 | 0.72 | 0.74 | 0.70 | 0.52 | 0.65 | 0.65 |
| DeepFloyd-IF (Saharia et al., 2022) | Diff. | 0.83 | 0.85 | 0.81 | 0.82 | 0.89 | 0.84 | 0.74 | 0.74 | 0.71 | 0.53 | 0.68 | 0.66 |
| Midjourney v6 | Diff. | 0.88 | 0.87 | 0.87 | 0.87 | 0.91 | 0.87 | 0.78 | 0.78 | 0.79 | 0.50 | 0.76 | 0.69 |
| DALL-E 3 (Betker et al., 2023) | Diff. | 0.91 | 0.90 | 0.92 | 0.89 | 0.91 | **0.90** | 0.82 | 0.78 | 0.82 | 0.48 | 0.80 | 0.70 |
| EMU3 (Wang et al., 2024) | AR | 0.78 | 0.81 | 0.77 | 0.78 | 0.87 | 0.78 | 0.69 | 0.62 | 0.70 | 0.45 | 0.69 | 0.60 |
| SEED-X (Ge et al., 2024) | AR+Diff. | 0.86 | 0.88 | 0.85 | 0.85 | 0.90 | 0.86 | 0.79 | 0.77 | 0.77 | 0.56 | 0.73 | 0.70 |
| LlamaGen (Sun et al., 2024a) | AR | 0.75 | 0.75 | 0.74 | 0.76 | 0.75 | 0.74 | 0.63 | 0.68 | 0.69 | 0.48 | 0.63 | 0.59 |
| LlamaGen-**Dexion** | AR | 0.75 | 0.77 | 0.73 | 0.76 | 0.79 | 0.74 | 0.68 | 0.67 | 0.72 | 0.67 | 0.72 | 0.68 |
| Janus-Pro (Chen et al., 2025) | AR | 0.87 | 0.88 | 0.87 | 0.87 | 0.91 | 0.86 | 0.77 | 0.78 | 0.76 | 0.42 | 0.72 | 0.66 |
| Janus-Pro-**Dexion** | AR | 0.87 | 0.89 | 0.87 | 0.89 | 0.92 | 0.87 | 0.79 | 0.79 | 0.76 | 0.57 | 0.77 | 0.74 |

Table 8: VQAScore evaluation of image generation on GenAI-Bench.

| Method | Type | # Params | Single Obj. | Two Obj. | Counting | Colors | Position | Color Attri. | Overall ↑ |
|---|---|---|---|---|---|---|---|---|---|
| LDM (Rombach et al., 2022) | Diff. | 1.4B | 0.92 | 0.29 | 0.23 | 0.70 | 0.02 | 0.05 | 0.37 |
| SDv1.5 (Rombach et al., 2022) | Diff. | 0.9B | 0.97 | 0.38 | 0.35 | 0.76 | 0.04 | 0.06 | 0.43 |
| PixArt-alpha (Chen et al., 2024) | Diff. | 0.6B | 0.98 | 0.50 | 0.44 | 0.80 | 0.08 | 0.07 | 0.48 |
| SDv2.1 (Rombach et al., 2022) | Diff. | 0.9B | 0.98 | 0.51 | 0.44 | 0.85 | 0.07 | 0.17 | 0.50 |
| DALL-E 2 (Ramesh et al., 2022) | Diff. | 6.5B | 0.94 | 0.66 | 0.49 | 0.77 | 0.10 | 0.19 | 0.52 |
| SDXL (Podell et al., 2023) | Diff. | 2.6B | 0.98 | 0.74 | 0.39 | 0.85 | 0.15 | 0.23 | 0.55 |
| SD3 (Esser et al., 2024) | Diff. | 2B | 0.98 | 0.74 | 0.63 | 0.67 | 0.34 | 0.36 | 0.62 |
| Show-o (Xie et al., 2024) | AR.+Diff. | 1.3B | 0.95 | 0.52 | 0.49 | 0.82 | 0.11 | 0.28 | 0.53 |
| SEED-X (Ge et al., 2024) | AR.+Diff. | 17B | 0.97 | 0.58 | 0.26 | 0.80 | 0.19 | 0.14 | 0.49 |
| Transfusion (Zhou et al., 2024) | AR.+Diff. | 7.3B | - | - | - | - | - | - | 0.63 |
| EMU3 (Wang et al., 2024) | AR. | 8B | - | - | - | - | - | - | 0.66 |
| EMU3-DPO (Wang et al., 2024) | AR. | 8B | - | - | - | - | - | - | 0.64 |
| Janus (Wu et al., 2024a) | AR. | 1.3B | 0.97 | 0.68 | 0.30 | 0.84 | 0.46 | 0.42 | 0.61 |
| Emu3-Gen (Wang et al., 2024) | AR. | 8B | 0.98 | 0.71 | 0.34 | 0.81 | 0.17 | 0.21 | 0.54 |
| Chameleon (Team, 2024) | AR. | 7B | - | - | - | - | - | - | 0.39 |
| LlamaGen (Sun et al., 2024a) | AR. | 0.8B | 0.71 | 0.34 | 0.21 | 0.58 | 0.07 | 0.04 | 0.32 |
| LlamaGen-**Dexion** | AR. | 0.8B | 0.86 | 0.33 | 0.28 | 0.62 | 0.07 | 0.02 | 0.36 |
| Janus-Pro (Chen et al., 2025) | AR | 7B | 0.98 | 0.88 | 0.59 | 0.92 | 0.79 | 0.65 | 0.80 |
| Janus-Pro-**Dexion** | AR | 7B | 0.98 | 0.80 | 0.60 | 0.94 | 0.80 | 0.68 | 0.81 |

Table 9: Evaluation on the GenEval (Ghosh et al., 2024) benchmark.

incorporated an adversarial loss with a PatchGAN discriminator. The perceptual loss weight was set to 1.0, while the discriminator loss weight was 0.5. The discriminator was introduced after 60,000 steps, and its loss was calculated using the hinge loss formulation.

After the initial training, we fine-tuned the decoder at a resolution of $512 \times 512$ to enhance the output quality for higher-resolution images. During this stage, we reduced the learning rate to $3e - 5$ for more stable convergence. The global batch size was adjusted to 96 to accommodate GPU memory constraints while maintaining training efficiency.

To fine-tune the multi-resolution decoder, we slightly increase the initial learning rate to $1e - 4$ and set the global batch size to 32, accounting for the presence of $1024 \times 1024$ images. The decoder is fine-tuned directly on our synthetic high-aesthetic-quality dataset.

### C.2 TRAINING DEXION REFINER

We trained all three variants of the Token Refiner using a consistent training setup to ensure robust and generalizable performance. The learning rate was set to $3e - 4$ with a global batch size of 128. We applied a weight decay of $0.045$ and optimized the model using the Adam optimizer with $\beta_1 = 0.9$ and $\beta_2 = 0.96$. To improve generalization, we incorporated label smoothing with a factor of 0.1. During training, the mask ratio was randomly sampled from the range $[0.001, 0.3)$, enabling the model to learn across varying levels of token masking. This setup allowed the Token Refiner to progressively infer masked tokens with high accuracy, leading to refined and coherent outputs across different masking scenarios.

## D GENERATION EVALUATION

In this section, we present detailed evaluation results as a supplement to Table 7. The comprehensive results for GenAI-Bench are shown in Table 8, while the results for GenEval are provided in Table 9.

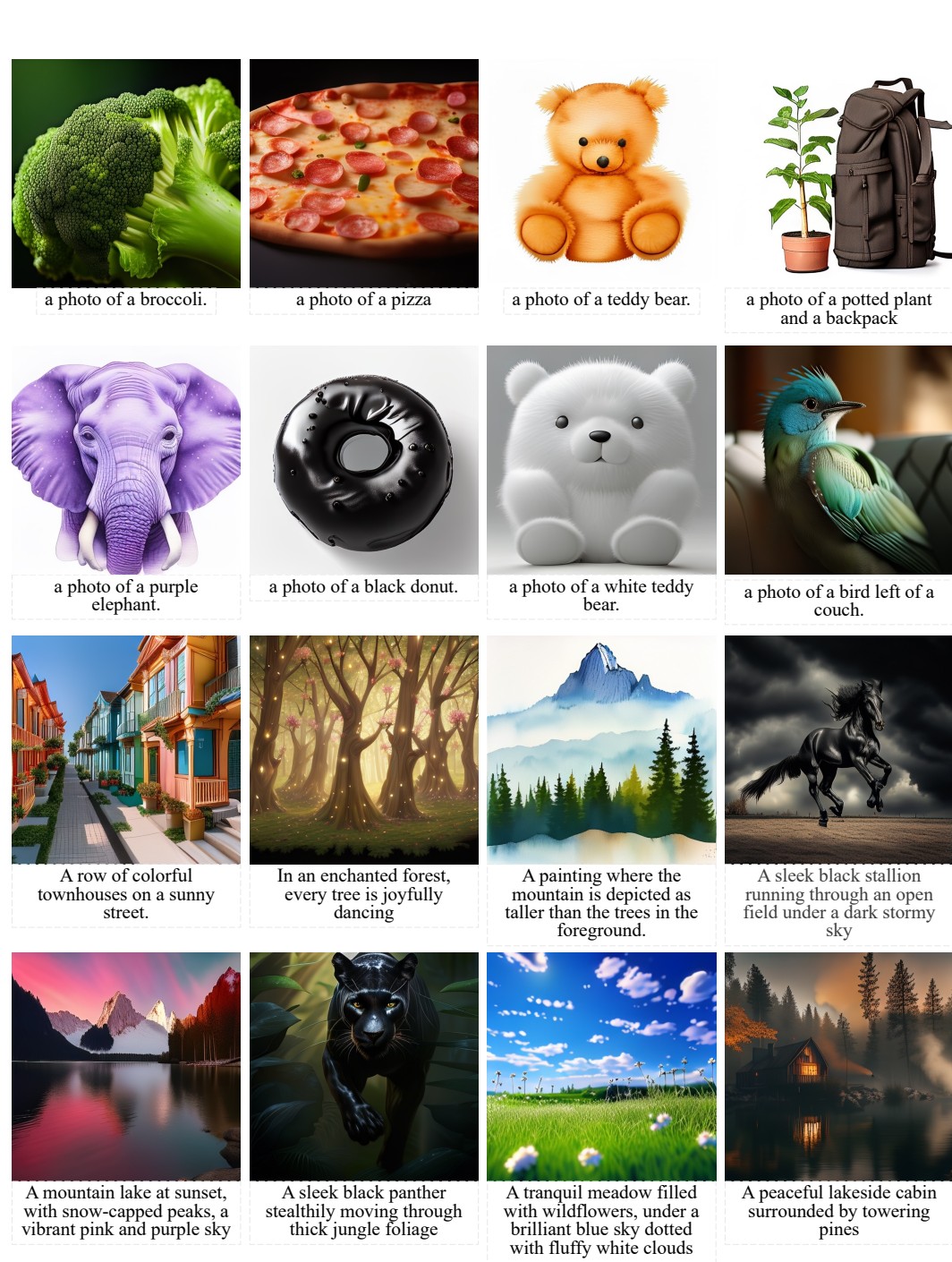

Figure 12: We show more examples generated by ARGen-Dexion (based on LlamaGen). All images are at a resolution of $512 \times 512$.

# E VISUAL GALLERY

**More Visual Examples** To better comprehensively illustrate the generation quality of our approach, we provide a diverse set of additional visual examples at a resolution of $512 \times 512$ in Fig. 12, each accompanied by the corresponding prompt used. With the augmentation of ARGen-Dexion, we can see the image quality is largely boosted.

**Effectiveness of Token Refiner** Leveraging global coherence, Token Refiner aims to subtly refine the tokens generated by the AR model without altering the overall context. Figure 13 illustrates the effectiveness of our proposed Token Refiner in ARGen-Dexion. Without global coherence, ARGen struggles to handle accumulated errors and fails to predict fine details accurately. By introducing our refiner, we observe a noticeable improvement in the refinement of details, such as the fox's eyes and the shining flowers. Additionally, the refiner is capable of subtly adjusting the global aesthetics, as seen in the gold lion image. In future work, we aim to design a more carefully tuned refiner to improve global structure within an image.

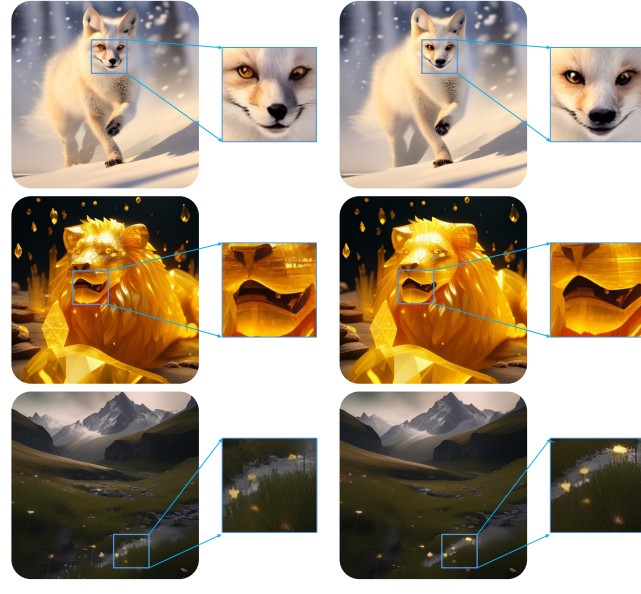

Without Refiner      With Refiner

Figure 13: Token Refiner excels at enhancing image details without significantly altering the original context, effectively addressing the intricacies that AR models often struggle to manage.

