# OpenReview forum: "ARGen-Dexion: Autoregressive Image Generation Made Stronger by Vision Decoder"
_ICLR.cc/2026/Conference — ICLR 2026 Conference Withdrawn Submission_

### Official Review · Reviewer_dXGZ · 2025-10-20

**Soundness:** 3
**Presentation:** 3
**Contribution:** 2
**Rating:** 4
**Confidence:** 4

**Summary:**

The paper proposes the ARGen-Dexion model, which improves the quality and flexibility of image generation by optimizing the visual decoder. Specifically, the paper first enhances the image reconstruction capability by fine-tuning the decoder. Then, it trains a Token Refiner with bi-directional attention to correct the image using global contextual information. Finally, the paper introduces an adaptive pooling layer into the decoder to support multi-resolution generation.

**Strengths:**

1. The motivation of the paper is good. The main defects of the autoregressive model are the information loss caused by quantization and the insufficient expression ability of casual attention.
2. The paper provides a thorough exploration of the decoder's scaling, offering valuable insights.
3. The method can enhance the generation performance of a baseline model by only training the visual decoder, without retraining the LLM.

**Weaknesses:**

1. The paper introduces an additional token refiner process, which increases the complexity and inference time of the generation process.
2. The visualization for the token refiner (Figure 13) is unconvincing, as it only appears to show changes in details without improving prompt alignment.
3. Another line of work (e.g., Tar [1]) combines AR and Diffusion models by first generating tokens with an AR model and then using them as a condition for a diffusion model to regenerate the image. The paper should compare its method with this approach to validate the advantages of the token refiner.

[1] Vision as a Dialect: Unifying Visual Understanding and Generation via Text-Aligned Representations. NeurIPS 2025

**Questions:**

The authors claim that casual inference is a drawback of autoregressive models, yet they use a MaskGIT-like bi-directional attention model to refine the results. Considering that MaskGIT can also be used directly for generation, why is it necessary to perform autoregressive generation first instead of using the MaskGIT approach directly? What are the advantages of this two-stage method compared to MaskGIT?

---

### Official Review · Reviewer_7bPP · 2025-10-31

**Soundness:** 3
**Presentation:** 3
**Contribution:** 3
**Rating:** 6
**Confidence:** 4

**Summary:**

This paper introduces ARGen-Dexion, an enhanced vision decoder system designed to mitigate the core limitations of traditional autoregressive generation (ARGen) models, such as LlamaGen. Specifically, ARGen-Dexion addresses information loss stemming from discrete tokenization and the lack of global coherence inherent in causal inference with three designs:
1) A scaled, high-fidelity main decoder trained with reconstruction, perceptual, and adversarial losses to improve visual fidelity.
2)  A bi-directional Transformer-based Token Refiner that post-processes the AR model's output tokens. It infuses global context to
refine the AR model outputs, bypassing the AR model's causal constraints.
3) a resolution-aware training strategy enabling seamless multi-resolution and multi-aspect-ratio synthesis.

Experiments demonstrate that ARGen-Dexion, when applied to pre-trained models like LlamaGen, improves generation quality.

**Strengths:**

- Modular design and functional decoupling of the system: The approach shifts the burden of generating fine details and global context onto the enhanced decoder. The Token Refiner effectively serves as an external refinement module, which utilizes post-processing to inject global context and correct local errors stemming from the causal nature of the AR model, thereby maintaining the integrity of the original generation architecture.

- Practical value: The proposed method is practical. It provides a path to improve the many existing, deployed, and "frozen" AR-based MLLMs without incurring the exorbitant cost of retraining them.

- The empirical validation is robust and comprehensive. The authors provide in-depth scaling studies for the main decoder and thorough ablation studies.

**Weaknesses:**

- The paper only emphasizes the training efficiency of not modifying the AR model. However, it miss the discussion of the inference cost. The method adds a large multi-step token refiner and a scaled-up main decoder. It is suggested that the author add a discussion about this issue.

**Questions:**

See Weaknesses.

---

### Official Review · Reviewer_ioZH · 2025-11-01

**Soundness:** 2
**Presentation:** 2
**Contribution:** 3
**Rating:** 4
**Confidence:** 3

**Summary:**

This paper proposes ARGen-Dexion, a new type of decoder architecture for image autoregressive models. It attributes the performance lower generation quality of AR models to the discretization loss and causal mask during the generation process. To tackle this two problems, the authors introduce a two-stage architecture for the decoder. Specifically, the generated token are first feed into a bi-directional refiner, which is trained with mask prediction loss. Then the refined tokens are feeded into the main decoder to form the final image. By incorporating an adaptive pooling layer, the proposed decoder can generate images in different resolution. Results show that the proposed architecture outperforms the original decoder of LlamaGen and Janus-Pro models.

**Strengths:**

1. This paper focuses on improving the decoder, which is a new perspective for AR image generation.
2. The idea of token refining is straightforward and effective.
3. Experiments show that the proposed decoder outperforms the original decoder of LlamaGen and Janus-Pro models.

**Weaknesses:**

1. It is unclear to me what is the fundamental difference between the main decoder and the original decoder. It seems that they both appear to be CNN architecture trained with the same loss. If there is no essential difference, I suggest the authors reorganize the method section to highlight the token refiner as the key component. If not so, it is better for the author to explicitly discuss the difference and include ablation studies for different components.
2. The authors identify the discretization error as one of the major issues of AR models in Sec 3.1. However, in my understanding, the proposed method doesn't solve this problem. This decoder is still a network that maps discrete tokens to continuous images, which is consistent with the original decoder.
3. In my opinion, the actual generation results of combining the new decoder with existing AR models (Tab.7) should be the most important results. I suggest the authors present these results at the beginning of the experiment section, then followed by a discussion of the decoder’s own metrics.

**Questions:**

1. Could the authors provide a more detailed comparison of the computational cost between the proposed decoder and the original decoder?

---

### Official Review · Reviewer_reUY · 2025-11-01

**Soundness:** 3
**Presentation:** 3
**Contribution:** 2
**Rating:** 4
**Confidence:** 4

**Summary:**

This paper identifies that the generative quality of auto-regressive (AR) models is fundamentally constrained by visual tokenization loss and the limitations of causal, next-token prediction. Rather than re-engineering the AR model itself, the authors propose ARGen-Dexion, a powerful, post-processing vision decoder designed to compensate for these upstream flaws. This decoder-centric framework makes three primary contributions: first, it scales up the vision decoder and fine-tunes it for high-fidelity reconstruction; second, it introduces a bi-directional Transformer-based "Token Refiner" that operates on the AR model's output, using global context to correct errors and overcome causal limitations; and third, it incorporates a resolution-aware training strategy with adaptive pooling to enable multi-resolution and multi-aspect-ratio synthesis from a fixed-resolution AR model.

**Strengths:**

1. The paper’s core strength is its premise: shifting the generative burden from the (flawed) AR model to a powerful, dedicated, and independent decoder. This is a clever, modular approach that avoids costly retraining of the multi-billion parameter base models.
2. The approach is validated with performance gains on established benchmarks, experiments based on LlamaGen and Janus-Pro strengthen the claim that the method can integrates with all MLLMs within the ”next-token-prediction” framework.
3. The thorough and systematic study on scaling the main decoder's size, cost, and data (Sec 4.2) is a valuable contribution.

**Weaknesses:**

1. The paper's quantitative results (Table 7) reveal inconsistent and model-dependent gains. While the method provides a clear boost to the weaker LlamaGen baseline (+0.4 on GenEval), the improvement on the stronger Janus-Pro model is marginal at best (only +0.1 on GenEval). This pattern suggests the decoder's effectiveness diminishes significantly when applied to more advanced AR models, calling into question the true significance and general applicability of the proposed method beyond engineering effort.
2. The writer argues that "training ARGen-Dexion is both data and computationally efficient compared to training or fine-tuning an AR model for image generation"(line 255), but detailed GPUs and time cost of training is missing in both main body and supplementary part of the paper.

**Questions:**

1. Can the authors comment on the marginal gains on Janus-Pro for GenEval (+0.1) and 'Basic' GenAI-Bench (+0.1) (also +0.0 on 'Basic' GenAI-Bench of LlamaGen-Dexion)?
2. To validate the "computationally efficient" claim, what was the total training cost (in GPU hours) for the final Dexion module (Main Decoder + Base Refiner)? Will the refiner and two-stage decoder increase inference time a lot compare to original VQGAN? What are the trade-offs in inference time or complexity of all proposed components?
3. If the authors use their 'curated set of high-aesthetic-quality images' to simply fine-tune the LlamaGen model, will they gain the performance improvement? The current experiment fails to rule out the possibility that the performance improvement stems from the 'curated set of high-aesthetic-quality images' rather than the Dexion module itself.

---

### Note · Authors · 2026-01-06

I have read and agree with the venue's withdrawal policy on behalf of myself and my co-authors.